# The relationship between latent inhibition, divergent thinking, and eyewitness memory: A study on attention to irrelevant stimuli

Ixone Badiola-Lekue, Naiara Arriola, Gabriel Rodríguez*

Facultad de Psicología, Universidad del País Vasco (UPV/EHU), San Sebastián, Gipuzkoa, Spain

* gabriel.rodriguez@ehu.eus

## Abstract

Latent inhibition is a retardation in learning about a stimulus due to its prior exposure without explicit consequences. It has been suggested that individuals who tend to show less latent inhibition possess a "leaky" attentional style, finding it difficult to inhibit the processing of irrelevant information, which would manifest as an ability to generate uncommon and creative ideas. In the present study, we tested this hypothesis within a new framework—the field of eyewitness memory—by investigating whether the degree of attenuated latent inhibition is associated with the inclusion of more peripheral (seemingly irrelevant) information in testimonies about a witnessed event. In an experiment involving 116 university students, the LI Group was pre-exposed without masking to a target auditory stimulus without consequences, while the CTRL Group did not receive this pre-exposure. Subsequently, both groups engaged in a learning phase where they had to learn the association between the target stimulus and a novel outcome. A latent inhibition effect was observed, with participants in the LI Group showing retardation in learning this association compared to the CTRL Group. Additionally, all participants completed the Alternative Uses Task to assess divergent thinking and provided eyewitness testimonies based on the viewing of a video of a criminal event. We confirmed the known relationship between attenuated latent inhibition and creativity, finding that the lesser the latent inhibition exhibited, the higher the performance on the Alternative Uses Task. Moreover, we found that a lower degree of latent inhibition was associated with a higher number of peripheral details included in the testimonies. These results are discussed in terms of the leaky attention hypothesis, and an alternative explanation based on cognitive flexibility. According to this, individuals exhibiting attenuated latent inhibition may have an intact capacity to ignore irrelevant stimuli but would be highly efficient at rapidly redirecting their attention when changes occur.

**Data availability statement:** https://osf.io/9m2ke/.

**Funding:** This research was supported by grants from the Spanish Ministerio de Ciencia, Innovación y Universidades (Grant No. PID2020-120215GB-I00) and Gobierno Vasco (Grant No. IT-1501-22). The funders had no role in study design, data collection and analysis, decision to publish, or preparation of the manuscript.

**Competing interests:** The authors have declared that no competing interests exist.

Latent inhibition is a phenomenon that was first investigated in experiments with non-human animals within the framework of Pavlovian conditioning [1–3]. In this domain, latent inhibition was defined as a retardation in the occurrence of the conditioned response (CR) when the conditioned stimulus (CS) is repeatedly exposed without consequences before being paired with the unconditioned stimulus (US). Research on latent inhibition in the field of Pavlovian conditioning has spread prolifically over decades and continues to be very active today [4–7]. Accumulating evidence has shown that this is a robust and reproducible phenomenon across a variety of procedures, sensory modalities, and animal species [3,8]. The generality of latent inhibition in humans, however, has, to some extent, been difficult to establish using conditioning procedures [9] and remains a matter of debate [10,11].

Contemporary procedures for studying latent inhibition in humans have developed away from the traditional use of CSs and USs and the consequent need to measure specific CRs [10]. Alternatively, a broader approach has been adopted, resorting to procedures in which experimental subjects learn a cueing relationship between a target stimulus and an outcome that follows or accompanies it. From this approach, fully compatible with the more particular framework of conditioning studies mentioned above, latent inhibition is defined as a retardation in learning the target-outcome relationship that occurs when the target has been previously exposed in the absence of explicit outcomes [12–15].

Although a number of potentially complementary mechanisms have been suggested as contributing to the occurrence of latent inhibition [2,16–23], the most widely accepted and researched explanation is an attentional one. According to this explanation, pre-exposure to the target in the absence of consequences allows learning that it does not signal anything important, which in turn leads to a loss of attention to it. Latent inhibition arises when this decrease in attention makes it difficult to learn later that the target signals the occurrence of a specific outcome [3,17,21,24].

From this attentional perspective, latent inhibition is a phenomenon that reflects two important cognitive abilities that are the result of the interaction between attention and learning. The first of these is the ability to filter out, or ignore, stimuli that have been learned to be irrelevant. The second is the flexible ability to pay attention again to those previously ignored stimuli when the conditions under which they are presented change. This flexibility would be the reason why latent inhibition manifests as a retardation in, and not as a total impediment of, learning. When an unexpected outcome appears after pre-exposure, attention flexibly returns to the target stimulus, allowing learning of the target-outcome relationship to occur, although with some delay [17,21].

Based on this attentional explanation, several lines of research have explored the existence of individual differences in latent inhibition and their possible causes. The hypothesis that has received the most attention is that individuals who show attenuated or no latent inhibition might have an attentional functioning characterized by lax or ineffective filtering of irrelevant information. In other words, these individuals would have difficulty in ceasing to attend to the target stimulus during pre-exposure, which would attenuate the retardation in their subsequent learning about the target-outcome

relationship [25,26]. This type of ineffective or incomplete filtering of irrelevant information has sometimes been referred to in the literature as *leaky attention* [27], with this term capturing the idea of a functioning in which "leaks" of irrelevant information are attended to and gain access to processing resources. Other terms that have been used to refer to this type of attentional functioning are *unfocused attention* [28,29] or *diffuse attention* [30], which denote results that are opposite to the restriction of information that characterizes selective focused attention.

Traditionally, leaky attention has been considered a mechanism involved in the cognition of highly creative individuals [31,32]. Leakage of irrelevant information, which would be effectively inhibited in other less creative individuals, would allow connecting distant ideas and concepts, thus facilitating the synthesis of infrequent, novel, and useful ideas [32–35]. Linking this idea with the consideration that leaky attention might attenuate latent inhibition, led to the prediction that creative people might show attenuated latent inhibition [31,32,36,37]. Several studies have revealed favourable evidence for this prediction, measuring creativity both through divergent thinking tasks [38] and self-reports of creative achievement [39–42].

In the experiment presented here, we took all of these considerations about individual differences in latent inhibition and applied them in a novel way to a field of study—*eyewitness memory*—where the study of individual differences is also prominent [43–46]. Specifically, we set out to examine whether the tendency to display attenuated latent inhibition could act as a marker of the type of information individuals include in their testimonies. We specifically focused on the distinction between *central* and *peripheral* information, a differentiation that has been widely studied in the field of eyewitness memory.

Although there are slightly different definitions of these two types of information [47–51], in general, central information is considered to refer to the essential elements for understanding the narrative thread of an event, such as the main characters, their actions, and the key objects with which they interact. Peripheral information, on the other hand, would include contextual details and aspects that are not essential for understanding the development of the action, or that are distant from the main events. Several studies have shown that central information tends to be remembered better than peripheral information [50–53]. This difference has been interpreted in attentional terms: central aspects, being more relevant, would receive more attention and this would facilitate their recall compared to peripheral details [54,55].

This attentional explanatory framework is particularly interesting in light of the notion that attenuated latent inhibition is a marker of leaky attention. Given that peripheral information is often considered irrelevant, and that individuals with leaky attention tend to process this type of information to a greater extent, one might expect attenuated latent inhibition to serve as a marker of a tendency to include these peripheral details in testimony.

To test these hypotheses, we designed a study that assessed latent inhibition, divergent thinking, and the characteristics of testimonies in an eyewitness memory task (see Fig 1). As a starting point of the study, we hypothesized that individuals with higher creative performance, measured through the Alternative Uses Task [56]—an established assessment of divergent thinking abilities —would exhibit attenuated latent inhibition, replicating previous findings [39,40].

A key feature of our experiment was the use of a new procedure for measuring latent inhibition, in which pre-exposure to the target stimulus was not accompanied by a masking task. In doing so, we were responding to criticisms raised by previous studies [10], which have suggested that procedures with masked pre-exposure, such as those used in previous research on the relationship between latent inhibition and creativity [39,40], may not have measured genuine latent inhibition effects, but rather other types of effects related to inattention, such as *learned irrelevance* [11]. The presence of a masking task in these previous studies opens the possibility that the loss of attention to the target stimulus would not have occurred spontaneously as a direct, and exclusive, result of the non-reinforced pre-exposure. Instead, the initial requirement to perform a masking task during the pre-exposure phase might have induced perceived stimulus irrelevance and subsequent inattention.

To address this question and ensure that we were measuring a genuine latent inhibition effect, we employed a between-subjects task, with two groups: experimental (LI Group) and control (CTRL Group) (see Fig 1). The task was presented in a video-game format. Both groups were initially instructed to pay attention to all stimuli that were to be presented

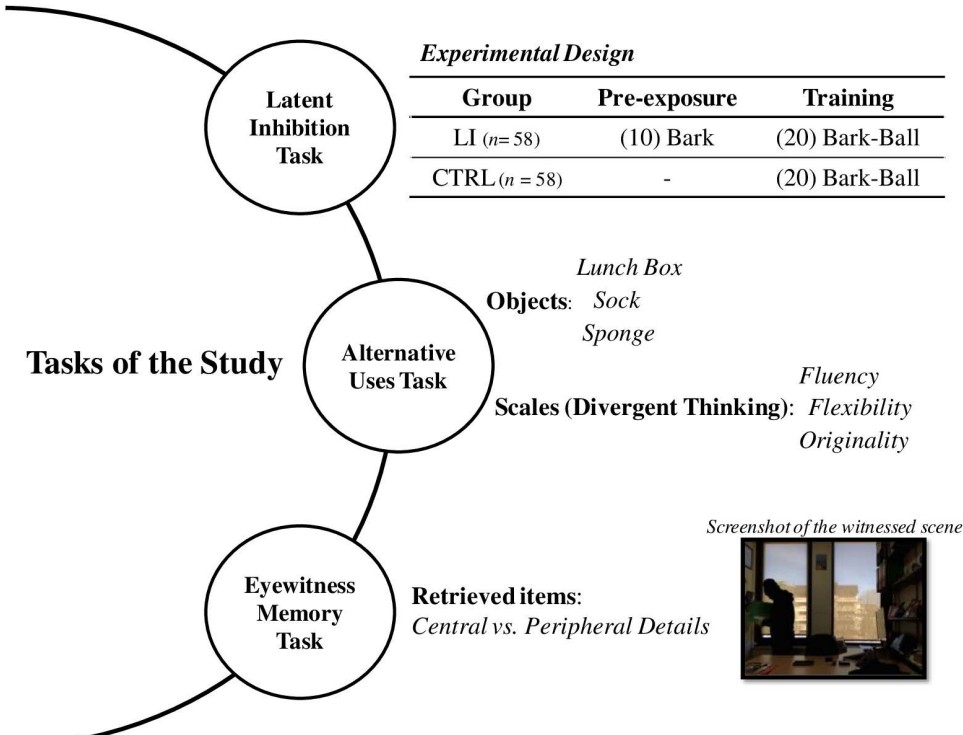

**Fig 1. Overview of the study tasks.** In the Latent Inhibition Task, both groups had to learn to bat the ball in the training phase. Successful batting was only possible if participants responded to a predictive stimulus (the bark of a dog) rather than to the visual appearance of the ball, as the latter did not provide enough time to react. The key difference between groups was that the LI group had previously received 10 isolated presentations of the bark without consequences during a pre-exposure phase, whereas the CTRL group did not receive this prior exposure. The Alternative Uses Task assessed divergent thinking through fluency, flexibility, and originality scores based on three common objects. Finally, the Eyewitness Memory Task involved recalling details from a witnessed scene, with retrieved items classified as central or peripheral details.

to them, indicating that any of them might be important later. This ensured that all stimuli were perceived, a priori, as potentially relevant, eliminating possible biases in directed attention.

After the instructions, a pre-exposure phase to the context began, during which only the LI Group was repeatedly exposed to a target auditory stimulus—the bark of a dog—without any relevant consequences. Afterwards, both groups participated in the training phase, in which participants had to learn to bat baseballs pitched at high speed. In order to successfully bat, participants had to anticipate the moment of batting by using the target stimulus (the bark) as a predictive cue for the appearance of the ball. The latent inhibition effect would be demonstrated if participants in the LI Group showed greater difficulty or delay in using the target stimulus as a predictive cue compared to those in the CTRL Group. The absence of a masking task, together with the use of instructions that matched the a priori relevance of all stimuli, would allow us to interpret with greater confidence that this effect would have been generated spontaneously by the characteristics of the non-reinforced pre-exposure.

For the eyewitness memory task, participants watched a video scene that simulated a conversation via Internet. The scene began after the supposed interlocutor momentarily left the office he/she was in. A few seconds later, a thief entered the office, searched it, and ended up stealing a folder (see a screenshot of the scene, as an example, in Fig 1). In this episode, the main actions of the thief and essential details about the objects he interacted with, such as the stolen folder, can be considered central information. On the other hand, peripheral information would include true but non-essential details to understand the event, such as objects with which the thief did not interact or other contextual aspects not directly related to the theft (e.g., whether the light was on or off).

If, as we hypothesized, leaky attention characterizes individuals with attenuated latent inhibition, we would expect these individuals to pay more attention to peripheral details and to include them more frequently in their testimonies. We did not have a clear starting prediction about the number of items of central information reported by individuals with attenuated latent inhibition. However, if these individuals pay more attention to irrelevant details, it seems reasonable to expect that this "distraction" would lead them to omit some central aspects of the event among those taken into account by individuals with intact latent inhibition.

## Method

### Participants

One hundred and sixteen students, 27 males (mean age = 18.2 years, SD = 0.4) and 89 females (mean age = 18.27 years, SD = 0.62), from the *University of the Basque Country* (UPV/EHU), volunteered to participate and gave their consent. They were randomly assigned to the LI (latent inhibition) group (n = 58; 13 males and 45 females) or to the CTRL (control) group (n = 58; 14 males and 44 females). All had normal or corrected-to-normal vision and normal hearing, and before the experiment began they were informed that their participation involved the performance of three cognitive tasks. They performed the tests individually in a quiet room via a standard computer. There were no statistically significant differences between the groups in terms of age, $t(114) = 1.15$, $p = 0.254$, and sex, $\chi^2(1) = 0.048$, $p = 0.826$. All study procedures were approved by the *Ethics Committee for Research Involving Human Subjects* (CEISH) of the UPV/EHU. All participants provided their consent both verbally and by accepting the conditions on the computer, which was required in order to begin their participation in the tasks. The recruitment period for the study extended from September 15, 2019, to December 1, 2019.

A power analysis was conducted using G*Power 3.1.9.7 [57] to estimate the minimum sample size required. The analysis indicated that the required sample size to achieve 80% power, as recommended by Cohen [58], for detecting a medium effect size ($f^2 = .15$) at a significance level of $\alpha = .05$ with one predictor in a two-tailed regression analysis was N = 55. We approximated this number by selecting a slightly larger sample size of $n = 58$ for each group in the latent inhibition experiment.

**Procedure.** Participants performed the tasks in this order: latent inhibition task, eyewitness memory task, alternative uses task, with a brief 2-min break between each. The tasks were administered in the same order for all participants. Overall, the testing session lasted approximately 40 minutes.

**Latent inhibition task.** When the task was initially presented to participants, they were told that it was designed in a video game format and required the use of headphones. When the participant accessed the task on the computer, the first thing he/she saw was a green screen with a button on the right with the word PLAY. The following instructions were presented on-screen (in Spanish): *Press the PLAY button when you are ready to enter the training field*. A typical video game-style pop song accompanied this presentation. When the participant pressed the button, the song stopped and the actual task began. The task consisted of two phases: the pre-exposure phase and the training phase in which the target-outcome relationship was learned.

*Pre-exposure phase*. As soon as this phase began, the following instructions (in Spanish) were presented on-screen for 6 seconds: *Pay attention to everything that happens... It might be useful later on*. The total duration of this exposure phase was 66 seconds. From the beginning, all participants were exposed to a park scene on the screen, featuring a mix of stationary items, like two trees and a fence in the foreground, and moving elements, such as girls playing football with a coach. This scene was accompanied by background sounds, including occasional distant human voices, birds singing, and other ambient noises appropriate to the setting.

During this phase, participants in the LI Group were also exposed to the target auditory stimulus, which was presented over the background sound. The target was a dog bark, 0.17 seconds in duration, which was presented 10 times. The first presentation occurred after ten seconds from the beginning of the phase, and the remaining presentations were delivered

at variable intervals, with the following values (in seconds) in random order: 13, 10, 9, 8, 7, 5, 4, 2, 0.5, and 0.5. Participants in the CTRL Group did not receive any presentation of this target sound during the pre-exposure phase.

*Training phase.* After a random variable interval of between 4 and 8 seconds from the end of the exposure phase, the training phase began in which the target-outcome relationship was learned. The phase change did not affect the background auditory environment, which continued to play to give continuity to the scene. All the visual elements presented on the screen in the pre-exposure phase also remained present, but now, in addition, a baseball player appeared in the centre of the screen. Below the player, the following instructions (in Spanish) appeared: *Now you can use the space bar to try to bat. You will get a point for each ball you manage to hit. Press the PLAY button when you are ready to start.* Along with these instructions, a visual representation of the computer keyboard also appeared with the silhouette of the space bar highlighted in red. From this point on, each time the participant pressed the space bar, the player-character would perform the typical swinging batting motion on the screen. To the right of this player also appeared a scoreboard, initially at zero, which subsequently increased as the player managed to hit balls during the training. When the participant pressed the PLAY button, the training began.

During the training a total of 20 balls were pitched to the player. The training was organized in 4 blocks of 5 balls each, but this division was not evident to the participants. The interval between the pitching of balls was variable and could take a random value between 5 and 9 seconds for each trial. When the participant succeeded in batting the ball, the contact of the bat with the ball was accompanied by a characteristic sound of wood being hit and the addition of a point on the scoreboard.

Critically, the pitching of the balls was programmed so that it was impossible to hit them based on their visual processing alone. When a participant pressed the space bar, a rapid transition of three consecutive images of the batter was initiated, creating the batting swing animation. Successful hits could only occur when the last image of the animation, in which the bat extended sufficiently to make contact with the ball, appeared. This image took 0.05 seconds to display after pressing the space bar. It took only 0.17 seconds for the ball to travel from its appearance at the bottom of the screen to the batting position. Since typical reaction times in simple motor tasks exceed this interval, hitting the balls based on their visual appearance was impossible. The only way to bat successfully was to anticipate the appearance of the ball by attending to the target stimulus—the bark of the dog—which occurred 0.4 seconds before the ball appeared. This allowed participants to press the space bar in advance, ensuring successful batting.

The 20 target-outcome (bark-ball) pairings in this phase allow learning of the signal relationship of the target to the outcome. This learning will become evident if the participant selectively responds (i.e., presses the space bar) only to the occurrence of the bark. To measure this behavioural manifestation of learning, the number of batting attempts and the number of successful bats (in which the ball was successfully hit) were recorded for each participant throughout the 4 training blocks. In each block, for each participant, a *batting success ratio* was calculated by dividing the number of successful hits by the total number of batting attempts in that block. The *overall batting success ratio* for the entire training was also calculated by dividing the total number of hits made during the entire training by the total number of attempts made. The latent inhibition effect would be manifested if participants in the LI group learn to respond to the bark with greater difficulty, and/or later, than participants in the CTRL group. This should be reflected in the observation of lower batting success ratios in the LI group relative to the CTRL group during training.

**Divergent thinking: Alternative uses task.** Divergent thinking was assessed using the alternative uses task [56,59]. At the beginning of the task, participants were given the following instructions (in Spanish) on the screen:

*Now we are going to present to you, one by one, three objects that we use on a daily basis with a certain frequency. The image of each object will be presented for TWO MINUTES. In that time you have to write in the text box situated under the image as many possible uses as you can think of for the object in question.*

*Here is an example. Imagine that the image of a "paper clip" appears. One might answer that the paper clip is used, for example: to gather papers; to organize cables on the desk; as a hairpin; as a skewer to pick up and eat olive; to prick a ball; to pinch and open a plastic package; as a bookmark.*

*Click with the mouse on the START button when you are ready to start the task.*

The objects presented were a lunch box, a sock, and a sponge. Three aspects of divergent thinking were assessed: *fluency* (number of responses generated), *flexibility* (number of different response categories; for example, if a person mentioned that the lunch box could be used as a *drum* and also as a *maraca* if filled with rice, two uses were computed in fluency and only one category in flexibility related to musical instruments), and *originality* (unusualness of responses, based on the statistical infrequency of each individual response within the current sample, with one point being awarded for each infrequent response that had been mentioned by less than 5% of the sample).

The fluency, flexibility, and originality scores were transformed into Z scores that were averaged to obtain a *divergent thinking score* (*DIV* score) for each subject.

**Eyewitness memory task.** The task started with the following instructions (in Spanish) being presented on the screen:

*Imagine that you are having a video conference with a friend. This person tells you that he/she is going to leave the room he/she is in for a few moments. While he/she is leaving, your camera is still on.*

*Press the space bar when you are ready to start playing the video.*

The video, which lasted 54 seconds, showed a person with his/her face covered entering a room and, after searching multiple locations and objects, finally taking a green folder with him/her (see an illustrative screenshot of the scene in Fig 1). Once the video was finished, with the following instructions (in Spanish), participants were asked to give their written testimony about the observed events, in a text box: *Please describe in as much detail as possible the scene you witnessed*.

There was no time limit for response, but all participants completed their response/testimony in less than 5 minutes.

Three judges reviewed the testimonies and identified a number of information items for which their probability of occurrence in the sample testimonies was calculated. Items included by more than 40% of the sample were considered *central information items*. Items included by less than 40% of the sample were considered *peripheral*. The information items in these categories are listed below. For ease of presentation, they are presented following the narrative thread of the scene.

**Central details.** Thirteen central information items were identified:

A person who looks like a man (68%), dressed in black (62%), with his face covered (67%), enters the room (87%) which looks like an office (66%), searches it (39%), checks a table (31%), checks some filing cabinets (38%), checks a first folder which he does not take (39%), checks a green folder (63%), takes that green folder (78%), puts it under his shirt/sweatshirt (63%), and leaves (86%).

**Peripheral details.** Thirty items of peripheral information were identified.

The thief appears to be a young adult (1%), of stocky build (1%), wearing a black cap (15%), black/dark trousers (2%), and a black sweatshirt (5%). He also wears sunglasses (2%) and latex gloves (1%).

Entering the room from the left (3%), stealthy (3%), the thief first turns on the light (9%). In the office there is a table (33%), and on the table there are magazines (6%), there is a mobile phone (9%) and money/purse (5%). In the room there is a window (38%) with the blind drawn (3%). From the window you can see buildings (13%), above the window there is a painting (3%), and nearby there is a ball (4%). The thief goes through papers under the window (17%).

In the office, on the right (14%), there are shelves (13%) and filing cabinets (16%). On the left there is a magazine rack (21%) with three folders (16%). The top folder is blue (3%) and the thief first takes this folder (1%). The middle folder is purple/red (3%) and the bottom one is green (3%). The folders have a rubber band closure (1%).

For each participant, the number of *central and peripheral items* included in his/her testimony was calculated. Also for each participant, an index indicating the proportion of peripheral information included in his testimony was calculated. This *peripheral/central index* was calculated by dividing the number of peripheral information items by the number of central information items included in each testimony. This index was computed with the objective of obtaining a measure

reflecting the relative frequency of the inclusion of peripheral information, independent of the total amount of information reported. This measure allows for a more accurate comparison of how participants distribute their attention and recall between central and peripheral details by controlling for possible individual differences in variables such as fluency and/or working memory capacity, which could influence the total amount of information reported.

Finally, the number of *untruthful information* items included in each testimony was also computed. The frequency of occurrence of these items was very low (only 6 false memories were found); so it was decided not to include this variable in the statistical analyses. In summary, therefore, the variables that were finally analysed in relation to the content of the testimonies were: the number of items of central information, the number of items of peripheral information, as well as the ratio index between peripheral information and central information.

**Statistical analysis.** To test our experimental hypotheses, we employed parametric tests (e.g., three-way ANOVA, Pearson's correlation, and hierarchical regression analyses). Prior to running these analyses using IBM SPSS Statistics 28.0.1.1, we conducted diagnostic checks to ensure that the underlying assumptions were met. Specifically, we assessed normality by examining skewness and kurtosis values (and through visual inspection of histograms, Q–Q plots, and residual plots). For some variables—such as the Originality scale from the Alternative Uses Task and the *peripheral/central index* from the Eyewitness Memory Task —skewness and kurtosis indicated moderate deviations from normality; however, given our sample size and the robust nature of parametric tests, these deviations are unlikely to compromise our results.

We also verified homogeneity of variances using Levene's Test. Furthermore, Mauchly's Test of Sphericity was used to assess the sphericity assumption for within-subject factors; when sphericity was violated, the Greenhouse–Geisser (or Huynh–Feldt) correction was applied to the degrees of freedom. Additionally, to evaluate multicollinearity among predictors in the regression models, we computed Variance Inflation Factors (VIFs) and tolerance values via the Regression procedure; all predictors yielded VIFs below 2 and tolerance values above 0.8, indicating acceptable levels of multicollinearity. Finally, independence of observations was ensured by the experimental design. These diagnostic steps confirm that our parametric tests are appropriately applied, with any minor deviations likely resulting in conservative estimates (i.e., increasing the risk of Type II error rather than false positives).

**Transparency and openness.** In this study, we detail how we determined our sample size, as well as all manipulations and measures used in the study. Statistical analyses were conducted using IBM SPSS Statistics 28.0.1.1. The power analysis was performed using G*Power 3.1.9.7. The design of the study and its analysis were not pre-registered. Data, along with illustrative examples of the tasks, are available on the Open Science Framework (https://osf.io/9m2ke/). Additional details, including the programming code for the tasks, are available upon request from the corresponding author (gabriel.rodriguez@ehu.eus).

# Results

## Alternative uses task

Table 1 presents the means, standard deviations, skewness, kurtosis, and correlations of the raw scores of the participants on the three scales of the Alternative Uses Task: fluency, flexibility and originality. Note that while fluency and

**Table 1. Descriptive Statistics and Correlations for Divergent Thinking Scales in the Alternative Uses Task.**

| Divergent Thinking Scale (n = 116) | Mean | Std. Deviation | Skewness | Kurtosis | 1 | 2 |
|---|---|---|---|---|---|---|
| 1. Fluency | 10.83 | 4.72 | 0.39 | 0.74 | – | |
| 2. Flexibility | 8.88 | 3.65 | 0.32 | 0.68 | .952** | – |
| 3. Originality | 1.8 | 1.85 | 1.59 | 3.14 | .617** | .541** |

*Note.* * The correlation is significant at a $p < 0.05$ level (2-tailed). ** The correlation is significant at a $p < 0.01$ level (2-tailed).

flexibility exhibited acceptable skewness (0.39 and 0.32, respectively) and kurtosis (0.74 and 0.68, respectively), the originality scale showed higher skewness (1.59) and kurtosis (3.14), reflecting a more peaked and heavy-tailed distribution. High positive correlations were observed between the scores of these three scales, suggesting that participants who generated a greater number of alternative uses (fluency) also tended to produce more diverse (flexibility) and infrequent (originality) ideas. However, it is noteworthy that, while the relationship between fluency and flexibility was very strong, the correlations between originality and the other two scales were weaker. This pattern of correlations supports the idea that the different dimensions of divergent thinking, although conceptually distinct, are related to each other and reflect integrated cognitive functioning in high-performing creative individuals.

### Eyewitness memory task

Table 2 presents the data corresponding to the testimonies obtained in the eyewitness memory task. Means, standard deviations, skewness, kurtosis, and correlations are presented for the number of central and peripheral information items, as well as for the peripheral/central index. While the Central information variable exhibits an acceptable distribution (skewness = −0.51; kurtosis = 0.14), the Peripheral variable shows moderate positive skewness (1.27) and kurtosis (1.25), and the Perif/Cent index demonstrates notable skewness (1.95) and kurtosis (5.66). As expected, the number of central information items included in the testimonies was greater than that of peripheral items. No significant correlation was observed between the number of central and peripheral items reported by the participants, suggesting that the encoding and/or recall of these two types of information may be mediated by relatively independent mechanisms.

### Relationship between the divergent thinking and the eyewitness memory task

To analyze possible relationships between performance on the alternative uses task and the eyewitness memory task, we used the DIV (divergent) index, calculated by averaging the Z scores of the three fluency, flexibility, and originality scales. Weak, non-significant positive correlations were observed between the DIV index and the number of central information items ($r = .103$, $p = .271$), between the DIV index and the number of peripheral information items ($r = .134$, $p = .151$), and between the DIV index and the peripheral/central index ($r = .128$, $p = .172$).

### Latent inhibition

As can be seen in the Fig 2, both groups showed a progressive increase in these scores, indicating that they progressively learned the cueing relationship between the target (i.e., the bark) and the outcome (i.e., the appearance of the balls) during the training phase of the task. However, the increase in these scores was slower in the LI Group compared to the CTRL Group, illustrating the latent inhibition effect. This pattern was confirmed by a three-way analysis of variance (ANOVA) with Group, Gender, and Block as factors. Given that Mauchly's test indicated a violation of sphericity for the factor Block ($p < .001$), the Greenhouse–Geisser correction was applied to the within-subjects effects. The analysis revealed that the main effects of Group, $F(1,112) = 9.32$, $p = 0.003$, $\eta^2 = 0.08$, and Block, $F(2.51, 280.87) = 24.91$, $p < .001$, $\eta^2 = .18$ (Greenhouse–Geisser corrected), were significant. However, the interaction Group × Block did not reach significance,

**Table 2. Eyewitness Memory Task. Descriptive Statistics and Correlations for Central and Peripheral Information Items and the Proportion Index of Peripheral to Central Information (Perif/Cent Index).**

| Type of Information (n = 116) | Mean | Std. Deviation | Skewness | Kurtosis | 1 | 2 |
|---|---|---|---|---|---|---|
| 1. Central | 7.78 | 1.91 | −0.51 | 0.14 | – | |
| 2. Peripheral | 2.75 | 2.82 | 1.27 | 1.25 | .139 | – |
| 3. Perif/Cent Index | 0.38 | 0.42 | 1.95 | 5.66 | −.228* | .845** |

*Note.* * The correlation is significant at a $p < 0.05$ level (2-tailed). ** The correlation is significant at a $p < 0.01$ level (2-tailed).

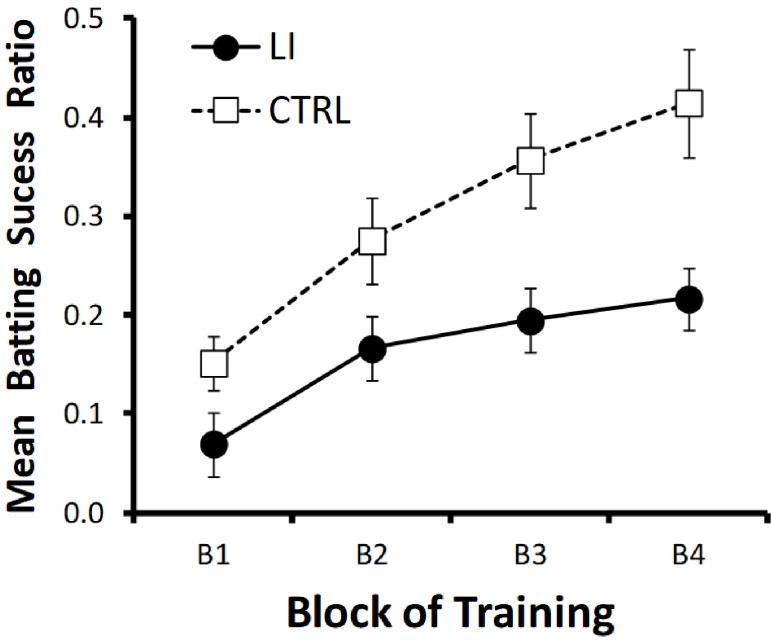

**Fig 2. Mean batting success ratios shown by the LI and CTRL groups across the four blocks of trials in the training phase of the LI task.** Both groups received 20 target-outcome (bark-ball) pairings during this training (5 pairings per training block). Prior to training, both groups were exposed to the training context. During this pre-exposure phase, the LI group also received 10 exposures to the target stimulus (i.e., the bark) in the absence of explicit consequences.

$F(2.51, 280.87) = 1.27$, p = .28. Neither the main effect of Gender, $F(1,112) = 2.69$, $p = 0.1$, nor any interaction involving this variable, $F$s < 1.43, $p$s > 0.23, was significant.

### Analysis of possible trivial differences between groups

Given the random assignment of participants to the LI and CTRL groups, we did not expect to find significant differences between them on variables derived from the alternative uses task and the eyewitness memory task. However, verifying the absence of these differences was important to rule out spurious explanations for other possible effects. To that end, Student t-tests were conducted to compare means between groups on the following variables: fluency (LI: M = 11.15, SD = 4.47; CTRL: M = 10.5, SD = 4.97), flexibility (LI: M = 9.1, SD = 3.42; CTRL: M = 8.65, SD = 3.87), originality (LI: M = 1.86, SD = 2.02; CTRL: M = 1.74, SD = 1.68), overall DIV score (LI: M = 11.15, SD = 4.47; CTRL: M = 10.5, SD = 4.97), central information (LI: M = 0.05, SD = 0.86; CTRL: M = −0.05, SD = 0.93), peripheral information (LI: M = 2.89, SD = 3.11; CTRL: M = 2.6, SD = 2.5), and ratio of peripheral to central information (LI: M = 0.42, SD = 0.5; CTRL: M = 0.32, SD = 0.3). No comparison showed significant differences between groups, $t$s(114) < 1.42, $p$s > 0.158.

### Relationship of latent inhibition with performance on the alternative uses task and the eyewitness memory tasks

According to the main hypothesis of this experiment, the effects of the non-reinforced pre-exposure received by the LI Group should generate a series of relationships between variables that will not occur in the non-pre-exposed CTRL Group. To verify these different patterns of results, correlations between variables from the three tasks of the study were analyzed separately in the two groups (see Table 3).

In the LI Group, a significant positive correlation was observed between the batting success ratio in the latent inhibition task and the DIV index from the Alternative Uses Task. This indicates that participants who performed better in the

**Table 3. Correlations between Batting Success Ratios in the Latent Inhibition Task, the DIV Index (Divergent Thinking) from the Alternative Uses Task, and Indicators of Information Types from the Eyewitness Memory Task.**

| | 1 | 2 | 3 | 4 |
|---|---|---|---|---|
| **Grupo LI (n = 58)** | | | | |
| 1. DIV | – | | | |
| 2. Central | .06 | – | | |
| 3. Peripheral | .2 | .12 | – | |
| 4. Perif/Cent Index | .2 | −.316 | .799** | – |
| 5. Batting Success Ratios | .305* | −.07 | .321* | .483* |
| **Grupo CTRL (n = 58)** | | | | |
| 1. DIV | – | | | |
| 2. Central | .17 | – | | |
| 3. Peripheral | .05 | .19 | – | |
| 4. Perif/Cent Index | .02 | −.01 | .968** | – |
| 5. Batting Success Ratios | .09 | −.03 | .04 | .03 |

*Note.* * The correlation is significant at a $p < 0.05$ level (2-tailed). ** The correlation is significant at a $p < 0.01$ level (2-tailed).

divergent thinking task tended to show a milder effect of the non-reinforced pre-exposure in disrupting the learning of the target-outcome relationship (i.e., tended to show attenuated latent inhibition). This interpretation, focused on the role of non-reinforced pre-exposure, is further supported by the lack of a significant correlation between the same variables in the CTRL Group, which did not receive that pre-exposure to the target.

Furthermore, in the LI Group, the batting success ratio was positively and significantly correlated with both the number of peripheral items reported in the testimonies of the eyewitness memory task and with the peripheral/central index. Notably, no significant correlations were observed between these same variables in the CTRL Group, indicating that the observed relationships are specific to the effect of unreinforced pre-exposure in the LI Group.

## Regression analyses

To assess how both latent inhibition and divergent thinking abilities influence performance on the eyewitness memory task, several two-step hierarchical regression analyses were performed. In all these analyses, in Step 1, we included the overall batting success ratio on the latent inhibition task as the main predictor, and in Step 2, we added the DIV index to assess any additional explanatory power. These analyses were applied separately for each of the study groups (LI Group and CTRL Group) and for each of the following three dependent variables: number of central information items included in the testimony, number of peripheral information items, and the peripheral/central index.

## Central information

In the LI Group, the overall batting success ratio on the latent inhibition task did not significantly predict the number of central information items reported, $R^2 = 0.005$, $F(1, 56) = 0.28$, $p = 0.599$, and adding the DIV index did not improve the model, $\Delta R^2 = 0.008$, $F_{change} = 0.45$, $p = 0.503$. Similarly, in the CTRL Group, neither the batting success ratio, $F(1, 56) = 0.04$, $p = 0.845$, nor the DIV index, $F_{change} = 1.72$, $p = 0.196$, significantly predicted the number of central information items.

## Peripheral information and peripheral/central ratio index

However, for the LI Group, the overall batting success ratio on the latent inhibition task significantly predicted the number of peripheral information items reported, $R^2 = 0.074$, $p = 0.039$, suggesting that a higher batting success ratio (i.e.,

 

attenuated latent inhibition) is associated with recalling more peripheral details. Adding the DIV index did not significantly improve the model, $\Delta R^2 = 0.013$, $p = 0.382$.

When examining the peripheral/central index, the overall batting success ratio on the latent inhibition task was also a significant predictor in the LI Group, $R^2 = 0.081$, $p = 0.028$, indicating that higher batting success ratios are associated with a greater focus on peripheral over central information. Again, adding the DIV index did not significantly improve the model, $\Delta R^2 = 0.013$, $p = 0.312$. In contrast, in the CTRL Group, neither the overall batting success ratio nor the DIV index significantly predicted the number of peripheral information items or the peripheral/central ratio index, $Fs < 0.12$, $ps > .0.729$.

In summary, results from this set of regression analyses suggest that in the LI Group, attenuated LI (indicated by higher batting success ratios on the latent inhibition task) is related to greater attention to peripheral details in the eyewitness memory task, while divergent thinking performance did not contribute significantly to the predictions. In the CTRL Group, with no pre-exposure to the target stimulus, no significant relationships were observed between the predictors and the eyewitness memory variables. This supports the idea that the relationships observed in the LI Group are specific to the effect of non-reinforced pre-exposure.

## General discussion

The present study explored the relationship between latent inhibition, divergent thinking, and eyewitness memory, aiming to extend existing research on how individual differences in latent inhibition may serve as a marker of attention to irrelevant stimuli. We built upon a widely accepted theoretical framework suggesting that individuals with high creative performance exhibit *leaky attention*, whereby irrelevant information, typically inhibited in individuals with more restrictive attentional styles, gains access to processing resources. One finding that supports this hypothesis is the observation that highly creative individuals tend to exhibit attenuated or reduced latent inhibition [39]. The supposed leaky attention of highly creative individuals would make it difficult for them to fully ignore the target stimulus during its pre-exposure in the absence of consequences (i.e., when it is irrelevant), thereby resulting in less disruption to subsequent learning of the target-outcome relationship (i.e., attenuated latent inhibition).

Our results confirmed this well-established relationship between latent inhibition and creativity, as we found that the batting success ratio in the LI Group, but not in the CTRL Group, was positively associated with performance on the Alternative Uses Task, a validated measure of divergent thinking and creative performance [31,32,36]. A notable contribution of this finding is that it was obtained using a latent inhibition task with unmasked pre-exposure, addressing some criticisms regarding potential validity issues in latent inhibition tasks employed in previous studies [10].

The most innovative aspect of our study was the use of an additional strategy involving eyewitness memory to address whether the degree of latent inhibition exhibited by an individual can effectively serve as a marker of the amount of attention paid to irrelevant stimuli. More specifically, we investigated the relationship between the degree of latent inhibition demonstrated by participants and the amount of central (or relevant) and peripheral (apparently irrelevant) information included in their testimonies about a witnessed event. In line with the leaky attention hypothesis, we found that lower levels of latent inhibition were associated with a greater tendency to include peripheral details in testimonies. This finding supports the idea that individuals with reduced latent inhibition tend to allocate more attention to seemingly irrelevant stimuli.

This result is relevant for several reasons. First, although this study has a primarily theoretical focus, our findings suggest the potential for developing an applied line of research. Latent inhibition could be exploited as a cognitive marker to predict the characteristics of testimonies in judicial contexts, which would be of great value to practitioners in this field.

Secondly, from a more basic and less applied perspective, the observed relationship between latent inhibition and attention to peripheral details in the eyewitness memory task provides additional insights into the mechanisms involved in latent inhibition and different cognitive processing styles. Our findings, like previous studies linking reduced latent

inhibition to creative performance [39], are consistent with the leaky attention hypothesis but do not constitute definitive evidence for it.

At least one alternative mechanism, distinct from a leaky attentional functioning, could also explain why individuals with reduced latent inhibition tend to show fewer signs of inattention to irrelevant stimuli: individual differences in cognitive flexibility. Instead of maintaining a constant leaky attention, individuals with attenuated latent inhibition might effectively ignore irrelevant stimuli while also being able to quickly restore attention to them when they perceive a significant change in context (e.g., the appearance of a new outcome signaled by the target stimulus). The availability of this flexibility to dynamically adjust attention could explain not only the ability to exhibit attenuated latent inhibition but also, as observed in our study, an enhanced ability to pay attention to peripheral details of a situation.

Two arguments support this alternative explanation based on the notion of flexibility. The first derives from a more detailed analysis of the explanation for attenuated latent inhibition in terms of the leaky attention hypothesis. According to this hypothesis, individuals with reduced latent inhibition struggle to inhibit attention to irrelevant stimuli. This assumption predicts that attention to the target stimulus during the target-outcome pairing phase will be relatively high. However, by the same logic, it would also be expected that leaky attention would continue allowing other irrelevant stimuli present during this phase to access controlled processing resources. This effect of leaky attention would hinder the establishment and behavioral expression of the target-outcome association, which is precisely the opposite of what is observed when latent inhibition is attenuated (i.e., facilitation of learning this association). For attenuated latent inhibition to be predicted by the leaky attention hypothesis, it must therefore be assumed that the effects of this type of attention are not constant, occurring during pre-exposure but fading or becoming less evident in the subsequent phase. This does not seem particularly plausible, and it would become entirely inconsistent in the context of an explanation that assumes leaky attention as a trait, or a relatively permanent attentional style [32,36,60]. Under these considerations, the leaky attention hypothesis seems less compelling compared to the flexibility-based explanation, which attributes decreased latent inhibition to enhanced flexibility as a stable trait.

The second and final argument supporting this alternative explanation based on flexibility is the well-established relationship between flexibility and creativity. Several studies have shown that creative individuals tend to exhibit reduced latent inhibition [39] and greater cognitive flexibility [27,61]. This alignment between creativity, flexibility, and attenuated latent inhibition not only reinforces the plausibility of the flexibility-based explanation but also provides a logical foundation for interpreting our findings within the broader framework of cognitive styles.

Although our results provide valuable information to improve our knowledge of the factors that influence the learning of inattention to irrelevant stimuli, it is necessary to recognize some limitations of our study. Firstly, our sample consisted solely of university students, with a relatively small number of male participants, which could limit the generalization of the findings to other age groups and more diverse populations. Secondly, as in almost all previous studies that have evaluated individual differences in latent inhibition [10], the measurement of the trait related to this phenomenon was based on performance in a single task. Although employing multiple measures could potentially increase the reliability of the construct, doing so presents a significant methodological challenge. Incorporating more than one latent inhibition task might lead participants to detect the solution strategy in the first task and subsequently recognize the similar structure of later tasks, prompting them to apply the same rule rather than responding naturally. This could artificially improve performance in subsequent tasks, thereby compromising the internal validity of the measurements. Therefore, it is essential that future studies explore alternative latent inhibition tasks that, despite measuring the same construct, present sufficient differences in their format and content to prevent participants from identifying common patterns. These limitations should be taken into account when interpreting our findings, and future research should address these aspects to improve both the external validity and the robustness of latent inhibition measurement. In summary, we believe that the findings presented in this work open new perspectives on how attenuated latent inhibition and the processing of irrelevant information are related to different attentional processing styles. Despite the aforementioned limitations, our results suggest that reduced latent

inhibition is associated with a greater focus on peripheral details in eyewitness memory. The incorporation of eyewitness memory as an additional type of test to investigate these relationships opens interesting lines of research. In recognition of our sample and measurement limitations, we emphasize that further studies are required to deepen our understanding of the mechanisms underlying variability in latent inhibition. For example, through a direct comparison of the roles of cognitive flexibility versus leaky attention, using diverse and non-overlapping measurement approaches.

## Author contributions

**Conceptualization:** Gabriel Rodríguez, Ixone Badiola-Lekue, Naiara Arriola.

**Data curation:** Ixone Badiola-Lekue, Naiara Arriola.

**Formal analysis:** Ixone Badiola-Lekue.

**Funding acquisition:** Gabriel Rodríguez.

**Methodology:** Gabriel Rodríguez, Ixone Badiola-Lekue, Naiara Arriola.

**Software:** Gabriel Rodríguez.

**Supervision:** Gabriel Rodríguez, Naiara Arriola.

**Writing – original draft:** Gabriel Rodríguez, Ixone Badiola-Lekue, Naiara Arriola.

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
