## [Decision Letter · Decision Letter 0]

16 Jan 2025

PONE-D-24-53417The Relationship between Latent Inhibition, Divergent Thinking, and Eyewitness Memory: A Study on Attention to Irrelevant StimuliPLOS ONE

Dear Dr. Rodriguez,

Thank you for submitting your manuscript to PLOS ONE. After careful consideration, we feel that it has merit but does not fully meet PLOS ONE’s publication criteria as it currently stands. Therefore, we invite you to submit a revised version of the manuscript that addresses the points raised during the review process.

The article is well-structured; however, a revision is needed that takes into account the reviewers' comments, with particular attention to the methodological aspect. In fact, it is advisable to enhance the clarity of the methodology used by providing a detailed description of the applied methods and the statistical analysis employed.

We look forward to receiving your revised manuscript.

Kind regards,

Andrea Cioffi

Academic Editor

PLOS ONE

“This research was supported by grants from the Spanish Ministerio de Ciencia, Innovación y Universidades (Grant No. PID2020-120215GB-I00) and Gobierno Vasco (Grant No. IT-1501-22).”

3. We note that Figure 1 in your submission contain copyrighted images. All PLOS content is published under the Creative Commons Attribution License (CC BY 4.0), which means that the manuscript, images, and Supporting Information files will be freely available online, and any third party is permitted to access, download, copy, distribute, and use these materials in any way, even commercially, with proper attribution. For more information, see our copyright guidelines: http://journals.plos.org/plosone/s/licenses-and-copyright.

Reviewers' comments:

Reviewer's Responses to Questions

**Comments to the Author**

1. Is the manuscript technically sound, and do the data support the conclusions?

Reviewer #1: Yes

Reviewer #2: Yes

2. Has the statistical analysis been performed appropriately and rigorously? 

Reviewer #1: No

Reviewer #2: Yes

3. Have the authors made all data underlying the findings in their manuscript fully available?

Reviewer #1: Yes

Reviewer #2: Yes

4. Is the manuscript presented in an intelligible fashion and written in standard English?

Reviewer #1: Yes

Reviewer #2: Yes

5. Review Comments to the Author

Reviewer #1: The methods section provided important details regarding the sample size determination using G*Power and described the software used for the analysis, i.e., SPSS. However, a major limitation is the lack of explicit mention of the statistical analyses performed:

The use of G*Power to justify the sample size demonstrates methodological rigor. However, the absence of detailed information about the statistical tests conducted (e.g., ANOVA, regression, or correlation analyses) weakens the transparency of the study.

Readers cannot assess the exact and appropriateness of the statistical analysis used or their alignment with the research questions.

More detailed descriptions of how the software was used, beyond sample size determination, would enhance clarity.

Reviewer #2: The introduction section is written extremely well and sets out a clear and convincing argument to both contextualise and support the aims of the study.

The design, sample, procedure, and instruments are explained very well, with careful attention to detail and all technical aspects discussed clearly and in-depth. The methodological approach is novel and contributes meaningfully to advancing further empirical research in the field, and the description is highly suitable to support replication. Ethical requirements for the study were adhered to and are discussed briefly but sufficiently.

It would be useful to have more detail about the analytic approach adopted in the study, particularly about how the assumptions for the statistical techniques were checked and managed during the analytic process. For example, the skewness and kurtosis estimates in Tables 1 and 2 suggest that certain variables were not normally distributed in the sample – it would be useful to know how this was accounted for in the later analyses. Including a brief discussion about independence, normality, heteroscedasticity, and no multicollinearity for both the 3-way ANOVAs and the multiple regression models would also be useful, especially as these techniques provide the basis for the main conclusions drawn in the study. The results themselves are reported clearly and the interpretations made are accurate in relation to the analyses that were run, however it would be important to support these by verifying the suitability of the techniques based on meeting the assumptions for these.

The general discussion of the results is presented very well and sets out technically accurate, interesting, and comprehensive accounts of the findings. It would, however, be useful to include additional in-text citations to support some of the statements made referencing existing theory or hypotheses. At the end of the discussion, it would also be very useful to discuss the limitations of the study and to briefly contextualise the findings in light of these, as well as to slightly expand the discussion of future directions for research, given the novelty of the study and its potential for encouraging further research in the field.

Recommended technical changes:

Introduction, paragraph 1, line 3 – spelling of ‘inhibition’

Introduction, paragraph 4, lines 6-7 – wording slightly unclear, recommend re-phrasing (e.g. ‘…this flexibility would be the reason why latent inhibition manifests as a retardation in, and not as a total impediment of, learning…’)

General discussion, paragraph 7, lines 15-16 – wording slightly unclear, recommend re-phrasing (e.g. ‘…and it would become entirely inconsistent in the context of an explanation that assumes leaky attention as a trait…’)

6. PLOS authors have the option to publish the peer review history of their article (what does this mean? ). If published, this will include your full peer review and any attached files.

**Do you want your identity to be public for this peer review?** For information about this choice, including consent withdrawal, please see our Privacy Policy .

Reviewer #1: **Yes: ** ERNESTINA AIDOO

Reviewer #2: No

---

## [Author Response · Author response to Decision Letter 1]

6 Mar 2025

Dear Dr Cioffi:

We appreciate the opportunity to resubmit our manuscript titled "The Relationship between Latent Inhibition, Divergent Thinking, and Eyewitness Memory: A Study on Attention to Irrelevant Stimuli" for further consideration in PLOS ONE. Thank you and the reviewers for your work on its previous version. We have carefully addressed all the journal's requirements and have made the following revisions taking into account the reviewers' suggestions.

Journal Requirements

Formatting and File Naming Compliance:

We have revised the manuscript to ensure it meets PLOS ONE’s style requirements, including proper formatting and appropriate file naming conventions.

Funding Statement Update:

In accordance with PLOS ONE's guidelines, we have added a statement clarifying the role of the funders in our research.

Copyright Issues in Figure 1:

Due to incompatibilities between the CC-BY-SA 2.0 license from Scratch and PLOS ONE’s CC-BY 4.0 requirements, we have removed the original screenshots from the Latent Inhibition Task programmed in Scratch, as well as the images of objects from the Alternative Uses Task.

To address this, we have developed a new Figure 1, providing a schematic overview of the study's tasks, clearly illustrating the key elements and structure of each task without the use of copyrighted material.We have also updated the figure legend to clearly reflect these changes

Changes based on reviewers’ suggestions

Justification for the choice of our statistical analyses and provision of data supporting this choice

We appreciate the reviewers’ valuable feedback regarding the limited explanation of our statistical analyses. We acknowledge that the original manuscript did not fully describe the rationale for choosing the specific tests and how we verified their assumptions. In the revised manuscript, we have expanded the Methods and Results section to explicitly detail and support the analytical approach used.

Expansion of the discussion recognizing the limitations of the study and pointing out future directions for research

Following the recommendation of reviewer 2, we have added a new paragraph in the Discussion in which we analyze the limitations of our study and suggest avenues for future research.

Clarification and correction of some details in the description of the Latent Inhibition Task

We realized that the previous manuscript contained inaccuracies regarding the specific timing parameters critical for successful batting in the latent inhibition task. We previously reported the total time taken by the ball to cross the entire screen (0.3 s.) rather than the exact time from ball appearance to its arrival at the batting position (0.17 s.). Also, we did not explain clearly the relationship between the progress of the batting swing animation and the time structure of the trials. We have attempted to explain more clearly the logic of the task: the necessity for participants to respond based on the predictive auditory stimulus (the dog bark), occurring 0.4 seconds before the ball’s visual appearance, due to typical reaction times exceeding 0.17 seconds.

We have also corrected all minor errors and typos identified by the reviewers throughout the manuscript, thereby improving the precision and clarity of this revised version.

We believe that these revisions significantly improve the clarity, transparency and rigor of our manuscript. We thank you again for your time and work in reviewing our manuscript. We look forward to your comments.

Sincerely,

Gabriel Rodríguez

Universidad del País Vasco (UPV/EHU)

gabriel.rodriguez@ehu.eus

---

## [Decision Letter · Decision Letter 1]

21 May 2025

PONE-D-24-53417R1The Relationship between Latent Inhibition, Divergent Thinking, and Eyewitness Memory: A Study on Attention to Irrelevant StimuliPLOS ONE

Dear Dr. Rodriguez,

Thank you for submitting your manuscript to PLOS ONE. After careful consideration, we feel that it has merit but does not fully meet PLOS ONE’s publication criteria as it currently stands. Therefore, we invite you to submit a revised version of the manuscript that addresses the points raised during the review process.

Dear Authors,

Thank you for submitting your manuscript. After careful review, one of the reviewers found your work acceptable for publication, while another raised some issues that need your attention. Please review the comments from the reviewer and address them thoroughly.

We look forward to receiving your revised manuscript.

Kind regards,

Mario Treviño Villegas, Ph.D

Academic Editor

PLOS ONE

Journal Requirements:

Reviewers' comments:

Reviewer's Responses to Questions

**Comments to the Author**

1. If the authors have adequately addressed your comments raised in a previous round of review and you feel that this manuscript is now acceptable for publication, you may indicate that here to bypass the “Comments to the Author” section, enter your conflict of interest statement in the “Confidential to Editor” section, and submit your "Accept" recommendation.

Reviewer #1: All comments have been addressed

Reviewer #3: All comments have been addressed

2. Is the manuscript technically sound, and do the data support the conclusions?

Reviewer #1: Yes

Reviewer #3: Partly

3. Has the statistical analysis been performed appropriately and rigorously? 

Reviewer #1: Yes

Reviewer #3: No

4. Have the authors made all data underlying the findings in their manuscript fully available?

Reviewer #1: Yes

Reviewer #3: Yes

5. Is the manuscript presented in an intelligible fashion and written in standard English?

Reviewer #1: Yes

Reviewer #3: Yes

6. Review Comments to the Author

Reviewer #1: (No Response)

Reviewer #3: This study explores the relationship between latent inhibition and divergent thinking and attention. However, there are some deficiencies authors need address.

(1) The purpose of this study is not very clear, what deficiencies in this field at present? What are the problems need address? Which issues does this research intend to solve? These need to be strengthened.

(2) This study used two dependent variables, but what's the relationship between them? The overall logic of the research needs to be strengthened.

(3) Although the divergent thinking of three dimensions is very high, the natures of the dimensions are different, the author is not recommended to integrate them.

(4) The manipulation could be further checked; the manipulation of failure data should be cleaned.

(5) Data analysis should take two conditions as independent variables, rather than bating success rate.

(6) the gender, age, and other potential control variables should be considered when authors detect the difference between two group.

7. PLOS authors have the option to publish the peer review history of their article (what does this mean? ). If published, this will include your full peer review and any attached files.

**Do you want your identity to be public for this peer review?** For information about this choice, including consent withdrawal, please see our Privacy Policy .

Reviewer #1: **Yes: ** ERNESTINA AIDOO

Reviewer #3: **Yes: ** Quanlei Yu

---

## [Author Response · Author response to Decision Letter 2]

21 May 2025

Dear Dr. Treviño Villegas,

We are grateful for the time and effort devoted to reviewing our manuscript titled The Relationship between Latent Inhibition, Divergent Thinking, and Eyewitness Memory: A Study on Attention to Irrelevant Stimuli (manuscript ID: PONE-D-24-53417R1). We were pleased to see that Reviewer 1 considered all previous comments to have been adequately addressed and recommended the manuscript for publication.

Regarding Reviewer 3’s report, we respectfully submit that the concerns raised are already fully addressed in the current version of the manuscript. After carefully reviewing the six points raised, we found that each of them corresponds to content that is explicitly discussed in the Introduction, Method, Results, or Discussion sections. In fact, we believe that modifying the manuscript to restate these points would be redundant and risk making the text less clear or unnecessarily repetitive.

We would certainly be happy to revise the manuscript further if there were a constructive way to address the reviewer’s concerns more directly. However, at present, we are not sure how to respond more proactively without reiterating content that is already there. If, as the Academic Editor, you consider that any of the issues raised by Reviewer 3 should nevertheless be explicitly reiterated or emphasized in the manuscript, we would of course be willing to revise the text accordingly.

We provide below a point-by-point response to Reviewer 3, indicating where in the manuscript each of the issues is already covered. We hope that this clarifies the coherence and soundness of our study and confirms that the comments raised do not reflect substantive flaws in the work.

We remain at your disposal should you require any further clarification.

Kind regards,

Gabriel Rodríguez

Response to Reviewer 3

We thank Reviewer 3 for their comments. However, we respectfully note that all six concerns raised in their review are already addressed in detail in the current version of the manuscript. We provide a point-by-point response below.

(1) “The purpose of this study is not very clear, what deficiencies in this field at present? What are the problems need address?”

We respectfully disagree. The Introduction section (pp. 3–6) clearly outlines the motivation for the study, including the unresolved issue of whether attenuated latent inhibition can be a valid marker of leaky attention and how this relates to the processing of peripheral information in eyewitness memory. The limitations of previous research using masked pre-exposure procedures are also discussed in detail (p. 6), and our methodological improvement is clearly justified.

(2) “This study used two dependent variables, but what's the relationship between them? The overall logic of the research needs to be strengthened.”

The relationship between the two tasks (divergent thinking and eyewitness memory) is explicitly discussed both in the Introduction and the Discussion (pp. 5–6 and pp. 18–20), where we explain how both types of tasks are theoretically linked to the processing of irrelevant or peripheral information. We also provide clear rationale for analyzing them together in the context of individual differences in attention.

(3) “Although the divergent thinking of three dimensions is very high, the natures of the dimensions are different, the author is not recommended to integrate them.”

We are aware of the differences among fluency, flexibility, and originality, and this is acknowledged in the manuscript (p. 14). For this reason, we report descriptive statistics and intercorrelations for each dimension separately (Table 1), and only compute the composite DIV score for the purpose of certain correlational and regression analyses. Importantly, we also report the separate contributions of each dimension where appropriate (p. 15–16), ensuring transparency and granularity in our analyses.

(4) “The manipulation could be further checked; the manipulation of failure data should be cleaned.”

The manipulation is robust and clearly described. We explain in detail how latent inhibition was operationalized and verified through the analysis of the batting success ratios across blocks (pp. 15–16 and Figure 2), showing a clear group effect. The data cleaning procedures are also described (p. 8), and we note that no participants were excluded based on poor performance, as all were able to complete the tasks and showed learning curves.

(5) “Data analysis should take two conditions as independent variables, rather than bating success rate.”

This comment appears to misunderstand the analytic strategy. Group (LI vs. CTRL) is indeed treated as a between-subjects independent variable in several analyses, including the ANOVA on batting success (p. 15). However, to test our key hypotheses about individual differences in latent inhibition, we also analyze batting success ratio as a continuous predictor, particularly within the LI group where variation in the degree of latent inhibition is meaningful (p. 17–18). This approach is theoretically motivated and statistically appropriate.

(6) “The gender, age, and other potential control variables should be considered when authors detect the difference between two group.”

We did consider these variables. The groups did not differ in age or gender distribution (p. 8), and gender was included in the main ANOVA as a factor (p. 15), with no significant effects or interactions found. Thus, these variables were appropriately evaluated and ruled out as confounds.

In summary, we appreciate the reviewer’s engagement with our work. However, we respectfully submit that all six concerns are already substantively addressed in the manuscript. Repeating or emphasizing these aspects further would result in unnecessary redundancy and reduced readability. Should the editor consider that any of these points must be made more explicit in the manuscript despite our justifications here, we will be happy to revise accordingly.

---

## [Editor Report · Decision Letter 2]

23 May 2025

The Relationship between Latent Inhibition, Divergent Thinking, and Eyewitness Memory: A Study on Attention to Irrelevant Stimuli

PONE-D-24-53417R2

Dear Dr. Rodriguez,

We’re pleased to inform you that your manuscript has been judged scientifically suitable for publication and will be formally accepted for publication once it meets all outstanding technical requirements.

Kind regards,

Mario Treviño Villegas, Ph.D

Academic Editor

PLOS ONE
---

## [Editor Report · Acceptance letter]

PONE-D-24-53417R2

PLOS ONE

Dear Dr. Rodriguez,

I'm pleased to inform you that your manuscript has been deemed suitable for publication in PLOS ONE. Congratulations! Your manuscript is now being handed over to our production team.

Kind regards,

on behalf of

Dr. Mario Treviño Villegas

Academic Editor

PLOS ONE